# Global, regional, and national burden of kidney dysfunction-attributed ischemic heart disease from 1990 to 2021: A systematic analysis of the Global Burden of Disease study 2021

Huaipeng Zhang, Guoqing Li, Xiangbing Wang, Ying Zhang [ID]*

Department of Cardiology, Heze Municipal Hospital, Heze, Shandong, China

* zhyinglove@163.com

## Abstract

### Background

Ischemic heart disease (IHD) remains a leading cause of global morbidity and mortality, with kidney dysfunction (KD) emerging as a critical independent risk factor. This study aimed to provide a comprehensive analysis of the global burden and trends of KD-attributed IHD from 1990 to 2021.

### Methods

We analyzed data from the Global Burden of Disease (GBD) study 2021, covering 204 countries and territories from 1990 to 2021. We examined trends in mortality and disability-adjusted life years (DALYs) of KD-attributable IHD, with stratification by sex, age, and socio-demographic index (SDI). The association between burden and SDI was evaluated using Spearman's correlation coefficient, and a smoothed curve model was developed to analyze the stage of the association.

### Results

From 1990 to 2021, global deaths and DALYs from KD-attributable IHD increased by 22.4% and 18.7%, respectively, despite a decline in age-standardized rates (ASRs). Low- and low-middle SDI regions experienced the highest burden, with increasing ASRs, while high-SDI regions showed significant reductions. Males exhibited a higher burden, with pronounced age-specific increases in IHD burden post-50 years. Significant threshold effects were observed, with substantial declines in ASRs only in regions with SDI values above 0.7.

### Conclusions

The global burden of KD-attributable IHD is increasing, especially in low- and middle-income countries. Socioeconomic development plays a crucial role in

**Data availability statement:** All data used in this study are publicly available without restriction from the Global Burden of Disease (GBD) Results Tool hosted by the Institute for Health Metrics and Evaluation (IHME): http://ghdx.healthdata.org/gbd-results-tool. The minimal dataset used in this analysis can be obtained by selecting relevant parameters (e.g., location, year, cause, risk factor, metric, and age group) through the query tool.

**Funding:** The author(s) received no specific funding for this work.

**Competing interests:** The authors have declared that no competing interests exist.

mitigating the disease burden, underscoring the need for targeted interventions, including chronic kidney disease screening, lifestyle modification, and integrated cardiovascular and renal care.

## Introduction

Ischemic heart disease (IHD) is the leading global cause of death and disability, largely driven by atherosclerosis and its complications such as myocardial infarction and heart failure [1,2]. While traditional risk factors like hypertension and diabetes are well recognized, kidney dysfunction (KD) has recently emerged as a novel, independent contributor to IHD risk [3–6]. KD promotes cardiovascular pathology via systemic inflammation, endothelial dysfunction, calcium-phosphate imbalance, and uremic toxin accumulation [7,8]. Epidemiologic data indicate that even modest reductions in glomerular filtration rate (eGFR<60 mL/min/1.73m$^2$) increase IHD risk by 40–60% [9–11], while end-stage kidney disease confers up to a 30-fold higher cardiovascular mortality risk [9,12,13]. Recent community-based studies reinforce these relationships: one study found that moderate CKD was independently associated with sudden cardiac arrest (SCA), with each 10 mL·min$^{-1}$·1.73 m$^2$ decrement in eGFR below 90 linked to an 24% increase in SCA risk [14]. In dialysis-dependent patients, roughly one quarter of SCA events occurred during or within one hour after dialysis, and the incidence of SCA on dialysis days was nearly three times higher than expected, implicating dialysis timing and acute electrolyte/acid–base shifts as potentially modifiable triggers [15].

However, unlike high systolic blood pressure and elevated fasting plasma glucose—whose contributions to IHD burden have been extensively quantified through the Global Burden of Disease (GBD) framework [16,17]—KD remains under-investigated as an attributable risk factor in global analyses. Most prior studies focus on high-income countries, overlooking geographic heterogeneity—especially in low- and middle-income settings where CKD care is limited [18]. Moreover, the spatiotemporal patterns of KD-attributable IHD burden and their associations with sociodemographic development remain underexplored [1,19]. Sex- and age-specific disparities also warrant investigation, as men may face higher risks due to lifestyle factors [20], and aging populations further amplify the cardiorenal burden [21].

To address this gap, we leveraged data from the GBD 2021 study [1] to: (1) estimate global, regional, and national trends in IHD burden attributable to KD from 1990 to 2021; (2) evaluate disparities by age, sex, and socio-demographic index (SDI); and (3) explore the SDI-related patterns in KD-attributable IHD burden over time.

## Materials and methods

### Data source

This analysis utilized de-identified data from the GBD 2021, accessible through the Global Health Data Exchange (GHDx: http://ghdx.healthdata.org/). Specifically, all analytic inputs were obtained using the GBD Results Tool on the GHDx platform

by querying the cause name "Ischemic heart disease" and the exposure/risk "Kidney dysfunction." For each location, year (1990–2021), sex, and age group we retrieved Deaths and DALYs. The GBD 2021 database integrates multisource inputs, including vital registration systems, hospital records, population surveys (e.g., Demographic and Health Surveys), disease registries, and published cohort studies, covering 204 countries and territories from 1990 to 2021 [1]. Estimation pipelines for mortality and DALYs employed standardized algorithms detailed in GBD methodological publications. To ensure comparability across populations, all rates were age-standardized using the GBD reference population structure. Ethical approval for data collection and analysis was obtained from the University of Washington Institutional Review Board, with a waiver of informed consent granted for secondary analyses of de-identified datasets.

## Definitions

IHD is defined as disease of the coronary arteries, usually from atherosclerosis, leading to angina, myocardial infarction (MI) or ischaemic cardiomyopathy. MI is defined according to the Fourth Universal Definition of MI. Incidence is estimated for any MI, first-ever or recurrent. Stable angina is defined as reversible myocardial ischaemia brought on by activity or stress, with diagnosis based on clinical symptoms [22]. In accordance with the GBD 2021 Risk Factors Study methodology, KD was categorized into four mutually exclusive stages using biochemical and functional criteria: Stage 1–2 CKD: Characterized by albuminuria [albumin-to-creatinine ratio (ACR) >30 mg/g] concomitant with preserved glomerular filtration (eGFR $\geq$ 60 mL/min/1.73 m$^2$); Stage 3 CKD: Moderately impaired renal function (eGFR 30–59 mL/min/1.73 m$^2$); Stage 4 CKD: Severely impaired renal function (eGFR 15–29 mL/min/1.73 m$^2$); Stage 5 CKD: Kidney failure (eGFR < 15 mL/min/1.73 m$^2$). Individuals undergoing renal replacement therapy, including hemodialysis or transplantation, were systematically excluded from all classification strata to prevent confounding [19,23].

## Descriptive analysis

We compared the absolute numbers of DALYs and deaths, as well as their age-standardized rates (ASRs), attributable to KD-attributable IHD between 1990 and 2021. DALYs represent the total years of healthy life lost due to disease, from onset to death. All data are presented with 95% uncertainty intervals (UI). 95% UI reflects the range within which the true value is expected to lie, incorporating not only statistical variability (such as sampling error) but also the systematic uncertainties in the model assumptions, data sources, and input variables. The GBD study utilizes a probabilistic model to estimate health outcomes, and the UI accounts for all potential sources of uncertainty, including variations in the input data, modeling techniques, and the assumptions made during the estimation process. The 95% UI thus represents a range of possible values for a given health estimate, with 95% of possible estimates expected to fall within this interval. We report age-standardized rates for the 21 GBD world regions—groups of countries and territories that are geographically proximate and share similar epidemiological characteristics—and for 204 individual countries.

Further stratifications were made by sex and by 15 age groups (25–29 years, 30–34 years,... up to 95 + years), as well as by SDI. Because GBD 2021 does not include data on the burden of KD-attributed IHD in individuals younger than 25 years of age, our focus is primarily on those 25 years of age and older. We categorized countries and territories into five SDI levels (Low, Low-Middle, Middle, High-Middle, and High) to explore the relationship between disease burden and socioeconomic development. The SDI ranges from 0 to 1, with higher values indicating higher levels of socioeconomic and developmental progress, and vice versa.

## Statistical analysis

To evaluate temporal trends in ASRs from 1990 to 2021, we used a linear regression model to calculate the estimated annual percentage change (EAPC). Assuming a linear relationship between the natural logarithm of ASR and time, we applied the model: $Y = \alpha + \beta X + \varepsilon$, where Y represents ln(ASR), X denotes calendar year, $\varepsilon$ is the error term, and $\beta$ is the

regression coefficient. EAPC was computed using the formula $EAPC_{with\ 95\%CI} = 100 \times (exp(\beta) - 1)$. The corresponding 95% confidence intervals (CI) was then assessed; if it included 0, the trend was deemed stable. Otherwise, it was classified as increasing (EAPC and 95% CI > 0) or decreasing (EAPC and 95% CI < 0). We employed Spearman's correlation coefficients to gauge the strength and direction of the association between SDI and ASRs. To further clarify the relationship between disease burden and socioeconomic development, we constructed smoothed curve models to explore correlations between ASRs and SDI. All statistical analyses were performed using R (version 4.3.1), and a two-sided p-value < 0.05 was considered statistically significant.

## Results

### Global burden of KD-attributable IHD, 1990–2021

From 1990 to 2021, global DALYs and deaths due to KD-attributable IHD increased by 18.7% (95% UI: 15.3–21.6) and 22.4% (95% UI: 19.1–25.8), respectively. However, age-standardized mortality rate (ASMR) and age-standardized DALY rate (ASDR) exhibited declining trends, with EAPCs of −1.29 (95% UI: −1.34 to −1.25) and −1.45 (95% UI: −1.49 to −1.42), respectively (Table 1 and Fig 1). Low-middle SDI regions bore the highest burden, with ASDR and ASMR reaching 457.91 (95% UI: 331.56–585.79) and 23.18 (95% UI: 16.33–29.63) per 100,000 in 2021, accompanied by positive EAPCs of 0.09 (95% CI: 0.02–0.15) and 0.25 (95% CI: 0.16–0.33). Conversely, high-SDI regions demonstrated substantial reductions: ASDR decreased from 408.18 (95% UI: 295.18–515.70) in 1990 to 156.04 (95% UI: 112.49–196.26) in 2021 [EAPC = −3.36(95% CI: −3.50 to −3.22)], while ASMR declined markedly [EAPC = −3.48(95% CI: −3.60 to −3.36)] (Table 1, Fig 1).

### Sex- and age-specific patterns

Men consistently exhibited higher absolute DALY counts than women throughout the study period. Notably, female deaths predominated before 2005, but male fatalities surpassed females thereafter (Fig 1), indicating a pronounced male-predominant burden in later years. In 2021, global DALY and death counts escalated with age, showing sharp increases after 55 years (Fig 2A, 2B). Age-specific rates rose exponentially post-50 years for DALYs and post-60 years for mortality, with males consistently exceeding females across all age groups (S1 Table).

### Regional and national heterogeneity

Central Asia recorded the highest ASDR [923.22(95% UI: 693.27–1169.67)] and ASMR [34,654(95% UI: 25,494–44,226)] in 2021, whereas High-income Asia Pacific had the lowest ASDR and ASMR (Table 1, Figs 3A, 3C). Most regions displayed decreasing trends, yet East Asia and Southern Sub-Saharan Africa experienced rising burdens (EAPC >0.5) (Figs 3B, 3D). Nationally, Belarus reported the highest ASDR [989.63(95% UI: 695.23–1323.04)] and ASMR [60.99(95% UI: 41.89–80.92)], while San Marino presented the lowest ASDR [95% UI: 56.44 (35.31–82.31)] and ASMR [3.70 (95% UI: 2.30–5.56)] (S2 Table). Denmark achieved the steepest declines [EAPC in ASDR = −5.80(95% CI: −5.98 to −5.62); EAPC in ASMR = 95% CI: −5.71(−5.87 to −5.54)], contrasting with increasing burdens in Lesotho, Zimbabwe, and Kenya (EAPC >1.0) (S2 Table).

### SDI-based disparities in KD-attributable IHD burden

Spearman analysis revealed inverse associations between SDI and KD-attributable burden at regional (ρ = −0.165) for DALYs, (ρ = −0.283) for deaths; (P < 0.001) and national levels (ρ = −0.328) for DALYs, (ρ = −0.257) for deaths; (P < 0.001). Spline models identified threshold effects: ASRs declined steeply in high-SDI regions (SDI > 0.7 for regions; SDI > 0.6 for nations) (Fig 4).

Across SDI strata, high-SDI regions experienced the most substantial improvements, with ASDR and ASMR for KD-attributable IHD declining by over 60% from 1990 to 2021. These trends were supported by the most pronounced negative

Table 1. The global DALYs, deaths, and their estimated annual percentage changes of kidney dysfunction-attributed ischemic heart disease from 1990 to 2021.

| Location | DALYs | | | | | Deaths | | | | |
|---|---|---|---|---|---|---|---|---|---|---|
| | Number of cases, 1990 | ASR, 1990 | Number of cases, 2021 | ASR, 2021 | EAPC | Number of cases, 1990 | ASR, 1990 | Number of cases, 2021 | ASR, 2021 | EAPC |
| Global | 16229802 (11947344, 20312854) | 450.95 (328.61, 568.32) | 26134286 (18902485, 33382272) | 309.84 (222.64, 395.39) | −1.29 (−1.34, −1.25) | 828212 (594880, 1050788) | 26.53 (18.85, 33.72) | 1398574 (976437, 1778399) | 17.18 (11.94, 21.83) | −1.45 (−1.49, −1.42) |
| **SDI** | | | | | | | | | | |
| Low SDI | 802165 (571994, 1055119) | 373.32 (266.60, 489.32) | 1656537 (1149341, 2135341) | 348.43 (242.95, 450.84) | −0.19 (−0.28, −0.09) | 32150 (22882, 42292) | 18.46 (13.04, 24.13) | 70676 (49139, 91608) | 18.32 (12.75, 23.63) | 0.13 (−0.01, 0.26) |
| Low-middle SDI | 2753568 (1990737, 3556002) | 462.16 (331.59, 590.84) | 6389959 (4607035, 8202054) | 457.91 (331.56, 585.79) | 0.09 (0.02, 0.15) | 110261 (78770, 141436) | 22.52 (16.13, 29.05) | 281111 (201825, 360775) | 23.18 (16.33, 29.63) | 0.25 (0.16, 0.33) |
| Middle SDI | 3649435 (2673847, 4645355) | 390.74 (285.55, 492.98) | 8504374 (6113640, 11007058) | 335.79 (240.63, 432.57) | −0.37 (−0.44, −0.29) | 159277 (116144, 201595) | 21.62 (15.52, 27.31) | 427797 (297227, 550315) | 18.81 (12.90, 24.18) | −0.26 (−0.36, −0.16) |
| High-middle SDI | 4509444 (3231148, 5714455) | 513.46 (366.68, 645.46) | 6002845 (4217730, 7810465) | 310.47 (219.16, 403.75) | −1.86 (−2.11, −1.61) | 242140 (169421, 308978) | 31.87 (22.51, 40.57) | 369622 (252988, 480879) | 19.50 (13.41, 25.50) | −1.74 (−1.93, −1.54) |
| High SDI | 4492532 (3239142, 5675926) | 408.18 (295.18, 515.70) | 3555959 (2546007, 4533696) | 156.04 (112.49, 196.26) | −3.36 (−3.50, −3.22) | 283180 (198594, 364807) | 26.23 (18.51, 33.77) | 247958 (174436, 319419) | 9.68 (6.86, 12.47) | −3.48 (−3.60, −3.36) |
| **Region** | | | | | | | | | | |
| Andean Latin America | 37529 (26882, 49482) | 193.87 (138.57, 253.38) | 71678 (49005, 96270) | 123.79 (84.88, 166.41) | −1.65 (−2.03, −1.27) | 1941 (1368, 2553) | 11.20 (7.76, 14.80) | 4034 (2742, 5460) | 7.22 (4.90, 9.77) | −1.62 (−1.99, −1.25) |
| Australasia | 93969 (65791, 120151) | 407.52 (286.17, 521.91) | 61659 (42788, 79440) | 102.03 (71.35, 129.87) | −4.72 (−4.83, −4.60) | 5882 (4027, 7599) | 26.74 (18.26, 34.54) | 4791 (3282, 6244) | 7.32 (5.01, 9.52) | −4.40 (−4.49, −4.31) |
| Caribbean | 109610 (78423, 140103) | 445.85 (320.08, 568.34) | 155357 (110635, 201619) | 285.79 (202.86, 371.59) | −1.45 (−1.65, −1.24) | 5693 (4007, 7309) | 25.38 (17.84, 32.47) | 8262 (5751, 10602) | 14.90 (10.32, 19.16) | −1.76 (−1.96, −1.56) |
| Central Asia | 488745 (369217, 602756) | 1134.16 (854.18, 1400.60) | 657701 (504531, 826588) | 923.22 (693.27, 1169.67) | −1.17 (−1.48, −0.85) | 25659 (18859, 32019) | 65.65 (48.42, 81.85) | 34654 (25494, 44226) | 55.41 (39.65, 70.73) | −0.94 (−1.19, −0.68) |
| Central Europe | 862701 (616923, 1106702) | 628.13 (448.47, 801.57) | 711306 (505525, 906044) | 306.73 (219.43, 389.10) | −2.68 (−2.80, −2.57) | 47700 (32951, 61710) | 38.28 (26.66, 49.30) | 48512 (33634, 62900) | 20.21 (14.04, 26.07) | −2.37 (−2.47, −2.27) |
| Central Latin America | 303708 (223726, 382212) | 399.32 (293.29, 503.04) | 820793 (594787, 1040950) | 334.92 (242.10, 423.72) | −0.72 (−0.96, −0.47) | 15287 (10998, 19301) | 23.18 (16.40, 29.27) | 45644 (32384, 58290) | 19.31 (13.69, 24.74) | −0.67 (−0.92, −0.42) |
| Central Sub-Saharan Africa | 97944 (68997, 137166) | 489.99 (345.95, 674.23) | 207545 (144164, 285456) | 427.31 (300.05, 588.81) | −0.68 (−0.77, −0.60) | 3989 (2807, 5525) | 25.44 (17.62, 35.05) | 8635 (6044, 11942) | 22.78 (15.93, 31.67) | −0.59 (−0.68, −0.51) |
| East Asia | 1675913 (1167695, 2190953) | 238.26 (165.64, 308.75) | 4177539 (2743889, 5793525) | 216.27 (141.92, 299.43) | 0.38 (0.02, 0.75) | 76550 (52728, 100092) | 14.64 (10.15, 19.18) | 255701 (165772, 353484) | 14.65 (9.57, 20.11) | 0.77 (0.35, 1.19) |

*(Continued)*

Table 1. (Continued)

| Location | DALYs | | | | | Deaths | | | | |
|---|---|---|---|---|---|---|---|---|---|---|
| | Number of cases, 1990 | ASR, 1990 | Number of cases, 2021 | ASR, 2021 | EAPC | Number of cases, 1990 | ASR, 1990 | Number of cases, 2021 | ASR, 2021 | EAPC |
| Eastern Europe | 2296291 (1627245, 2947162) | 910.63 (646.41, 1161.65) | 2511499 (1759954, 3256112) | 713.26 (502.32, 922.97) | −1.31 (−1.80, −0.81) | 127439 (88114, 163653) | 56.27 (39.09, 71.70) | 154660 (108057, 199608) | 43.43 (30.43, 56.03) | −1.25 (−1.69, −0.81) |
| Eastern Sub-Saharan Africa | 112631 (73398, 153251) | 156.47 (101.78, 216.67) | 254605 (165456, 355169) | 157.94 (101.62, 221.51) | −0.13 (−0.21, −0.06) | 4410 (2839, 6140) | 7.57 (5.02, 10.35) | 10414 (6659, 14602) | 8.08 (5.22, 11.36) | 0.08 (0.01, 0.16) |
| High-income Asia Pacific | 324020 (213940, 421942) | 176.16 (116.96, 228.44) | 368996 (238188, 481523) | 68.17 (44.88, 88.24) | −2.98 (−3.14, −2.83) | 19440 (12481, 25347) | 11.57 (7.37, 15.18) | 27614 (17398, 37385) | 4.18 (2.67, 5.51) | −3.15 (−3.37, −2.93) |
| High-income North America | 1768239 (1274387, 2244433) | 488.46 (354.57, 618.27) | 1546106 (1124532, 1939526) | 222.81 (163.53, 278.25) | −2.88 (−3.05, −2.71) | 114941 (80211, 147711) | 31.24 (21.89, 40.00) | 105144 (74630, 134822) | 14.14 (10.09, 18.05) | −2.92 (−3.08, −2.75) |
| North Africa and Middle East | 1274004 (895125, 1663964) | 798.15 (552.52, 1040.18) | 2405729 (1610493, 3203899) | 563.50 (375.24, 753.37) | −1.15 (−1.19, −1.12) | 54644 (37718, 71382) | 41.37 (27.82, 54.06) | 111367 (73910, 149246) | 30.92 (20.45, 41.43) | −0.92 (−0.97, −0.87) |
| Oceania | 16459 (11581, 22654) | 562.50 (404.29, 765.24) | 41034 (29404, 56426) | 541.08 (387.44, 724.62) | −0.08 (−0.13, −0.02) | 588 (423, 806) | 26.88 (19.12, 36.01) | 1520 (1086, 2038) | 25.92 (18.48, 34.54) | −0.09 (−0.13, −0.05) |
| South Asia | 2761371 (1998165, 3611797) | 476.86 (342.01, 621.21) | 6992130 (5028092, 9145083) | 487.41 (347.92, 636.77) | 0.19 (0.08, 0.30) | 104158 (73890, 136099) | 22.18 (15.48, 28.87) | 303733 (215307, 396410) | 24.54 (16.87, 31.82) | 0.51 (0.36, 0.67) |
| Southeast Asia | 1039962 (756647, 1349343) | 418.00 (301.39, 539.90) | 2556043 (1830671, 3338052) | 403.86 (289.87, 521.75) | −0.11 (−0.17, −0.05) | 42506 (30226, 55028) | 20.82 (14.69, 26.76) | 112314 (81060, 144081) | 20.51 (14.70, 26.23) | −0.06 (−0.14, 0.03) |
| Southern Latin America | 122521 (83283, 158365) | 283.04 (193.13, 366.04) | 95173 (63858, 125061) | 106.90 (71.90, 140.86) | −2.81 (−2.96, −2.66) | 6948 (4554, 9071) | 17.41 (11.43, 22.82) | 5816 (3831, 7652) | 6.34 (4.20, 8.35) | −2.85 (−3.03, −2.68) |
| Southern Sub-Saharan Africa | 71758 (53400, 89967) | 277.36 (202.30, 346.76) | 160528 (122290, 201071) | 303.58 (227.50, 377.49) | 0.23 (−0.16, 0.63) | 3226 (2347, 4068) | 14.75 (10.64, 18.81) | 7406 (5526, 9252) | 16.84 (12.46, 21.41) | 0.35 (−0.03, 0.73) |
| Tropical Latin America | 357770 (272614, 439398) | 417.70 (315.60, 516.31) | 521221 (394062, 647216) | 203.08 (153.23, 253.10) | −2.27 (−2.35, −2.19) | 16196 (12032, 20113) | 22.31 (16.27, 28.12) | 25061 (18372, 31576) | 10.05 (7.32, 12.74) | −2.40 (−2.52, −2.28) |
| Western Europe | 2146519 (1563718, 2733558) | 364.01 (268.93, 459.96) | 1240739 (893509, 1574823) | 112.00 (80.77, 141.02) | −4.05 (−4.18, −3.92) | 138445 (98823, 179485) | 23.55 (16.69, 30.29) | 96138 (67450, 124263) | 7.70 (5.49, 9.86) | −3.84 (−3.95, −3.72) |
| Western Sub-Saharan Africa | 268135 (194209, 359390) | 341.64 (245.37, 455.97) | 576905 (415599, 756707) | 337.58 (244.99, 437.84) | −0.00 (−0.14, 0.13) | 12571 (9006, 16777) | 19.07 (13.71, 25.09) | 27155 (19669, 35566) | 19.52 (14.37, 25.78) | 0.13 (0.01, 0.25) |

DALYs: disability-adjusted life years; ASR: age-standardized rate; EAPC: estimated annual percentage change.

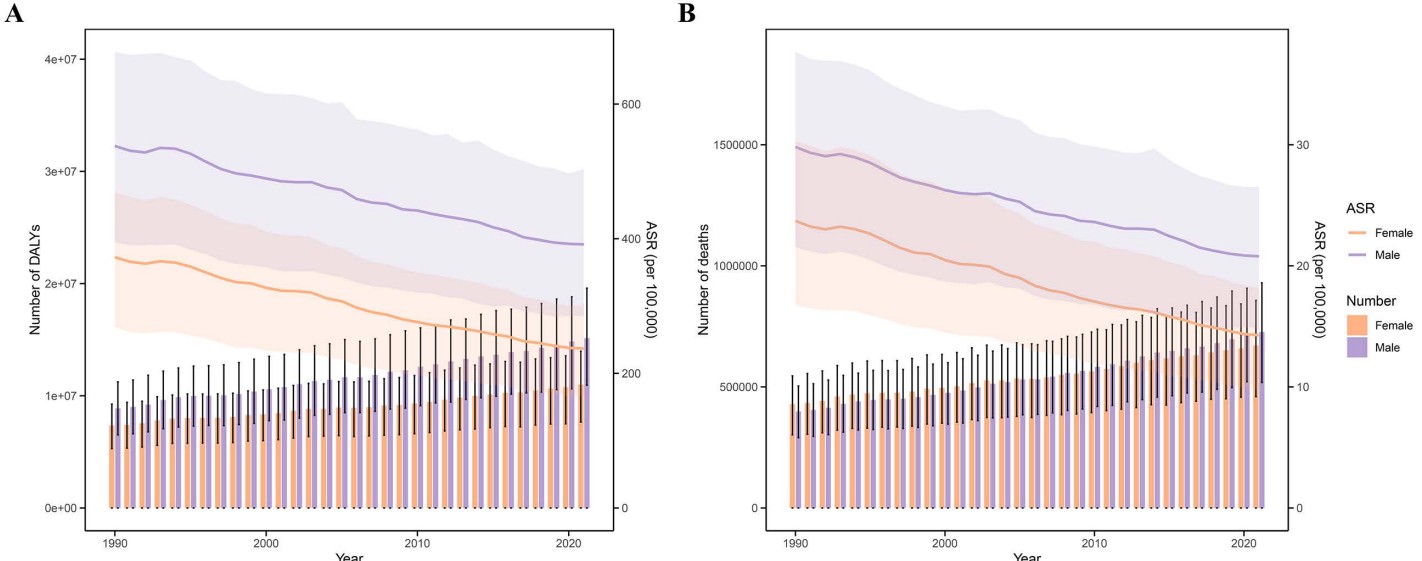

**Fig 1. Global trends in ischemic heart disease (IHD) burden attributable to kidney dysfunction (KD), 1990–2021. (A)** Global counts and age-standardized rates (ASRs) of disability-adjusted life years (DALYs) attributable to KD. **(B)** Global counts and ASRs of deaths from IHD attributable to KD. Error bars and shaded areas represent 95% uncertainty intervals. Abbreviations: KD, kidney dysfunction; IHD, ischemic heart disease; DALYs, disability-adjusted life years; ASR, age-standardized rate.

EAPCs, indicating successful reductions in disease burden likely driven by better CKD detection, cardiovascular risk control, and integrated healthcare systems. In contrast, low-middle SDI regions exhibited slight but significant increases in both ASDR and ASMR over the same period, reflected in small positive EAPCs. This suggests persistent healthcare access limitations and underdiagnosis of CKD in settings with fewer resources. Middle-SDI regions demonstrated relatively stable trends with minimal change in age-standardized rates.

## Discussion

### Research significance and key findings

This study provides the first comprehensive global analysis of the burden of IHD attributable to KD across 204 countries and territories from 1990 to 2021. By integrating data from the GBD 2021 study, we reveal critical epidemiological patterns shaped by aging populations, rising CKD prevalence, and persistent global health disparities. Three key findings emerged: (1) Despite declining ASRs, the absolute number of deaths and DALYs from KD-attributable IHD has risen significantly over the past three decades; (2) The burden is disproportionately high among older adults and men, with a notable shift toward male predominance post-2005; (3) Substantial disparities persist across regions and SDI levels, with low- and middle-SDI countries exhibiting rising trends, contrasting with marked declines in high-SDI regions. These findings underscore the growing impact of KD on cardiovascular health and the urgent need for integrated global prevention strategies.

### Pathophysiological mechanisms and age–sex differences

The rising prevalence of IHD is significantly influenced by the presence of KD, which serves as a key driver of adverse cardiac outcomes [9,10]. KD promotes the development of IHD through multiple interrelated pathophysiological pathways. These include activation of the renin–angiotensin–aldosterone system (RAAS), chronic systemic inflammation, oxidative stress, endothelial dysfunction, and disturbances in calcium-phosphate homeostasis—all of which contribute to vascular calcification, arterial stiffness, and accelerated coronary atherosclerosis, thereby impairing myocardial perfusion and

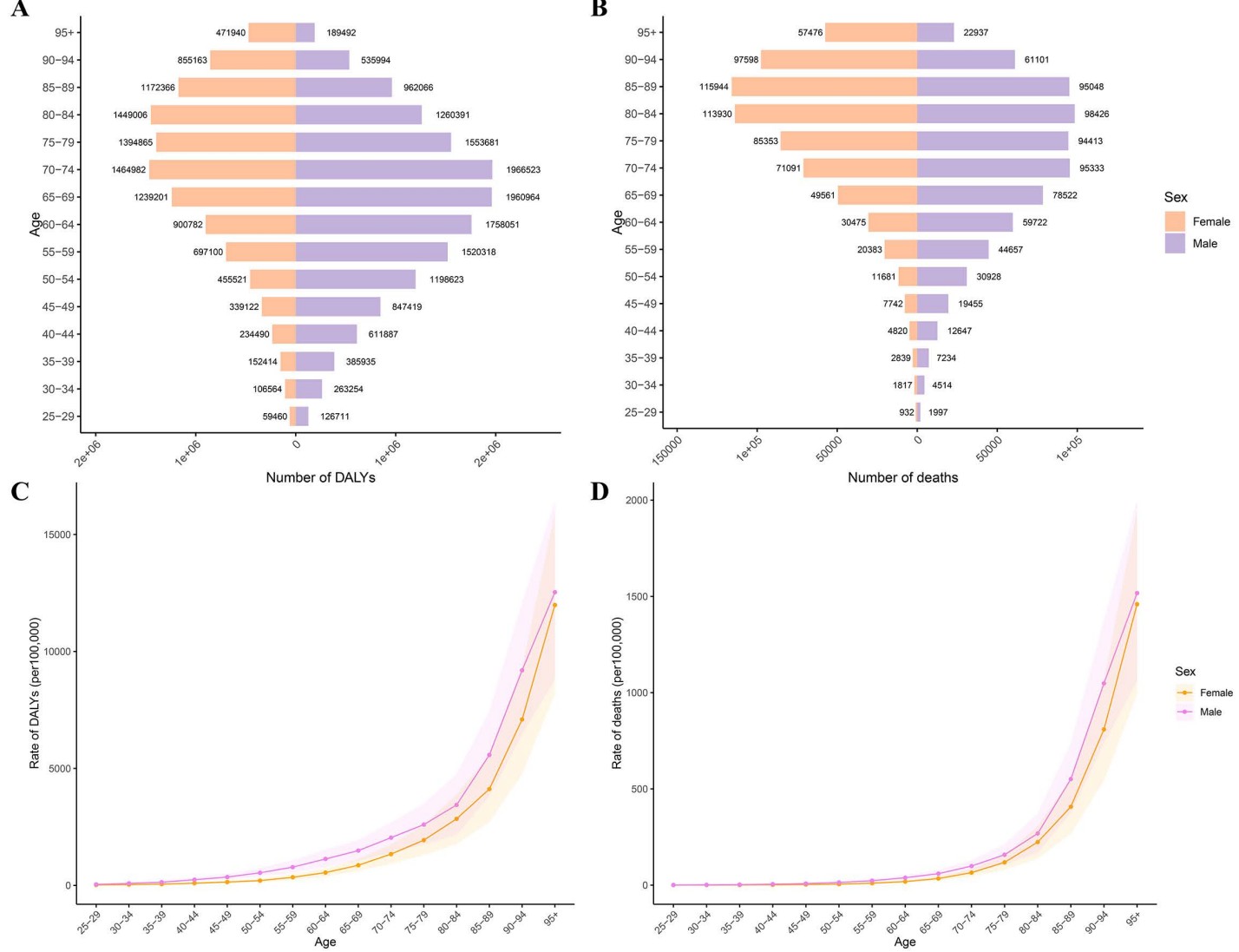

**Fig 2. Sex- and age-specific burden of ischemic heart disease (IHD) attributable to kidney dysfunction (KD) in 2021. (A, C)** Number and age-specific rates of disability-adjusted life years (DALYs) for males and females across 15 age groups. **(B, D)** Number and age-specific rates of deaths from IHD attributable to KD, stratified by sex and age. Males and older adults experienced higher absolute and relative burdens across all metrics. Abbreviations: KD, kidney dysfunction; IHD, ischemic heart disease; DALYs, disability-adjusted life years; ASR, age-standardized rate.

cardiac function [24]. Additionally, uremic toxins accumulate in patients with advanced CKD, exacerbating endothelial injury, promoting vascular remodeling, and altering drug pharmacokinetics, which complicates the management of IHD in this population [25].

The sex disparity we observed—men bearing a substantially larger KD-attributable IHD burden despite higher CKD prevalence in women—is consistent with prior syntheses and can be explained by several complementary factors. Part of the apparent female excess in CKD prevalence reflects demographic and measurement effects: women's longer life expectancy increases the chance of age-related eGFR decline, and creatinine-based eGFR equations can overestimate CKD in low-muscle-mass individuals [26]. In contrast, men more often progress rapidly to advanced CKD and accumulate cardiometabolic comorbidity (notably poorly controlled hypertension, diabetes, and higher smoking prevalence), which raises their incidence

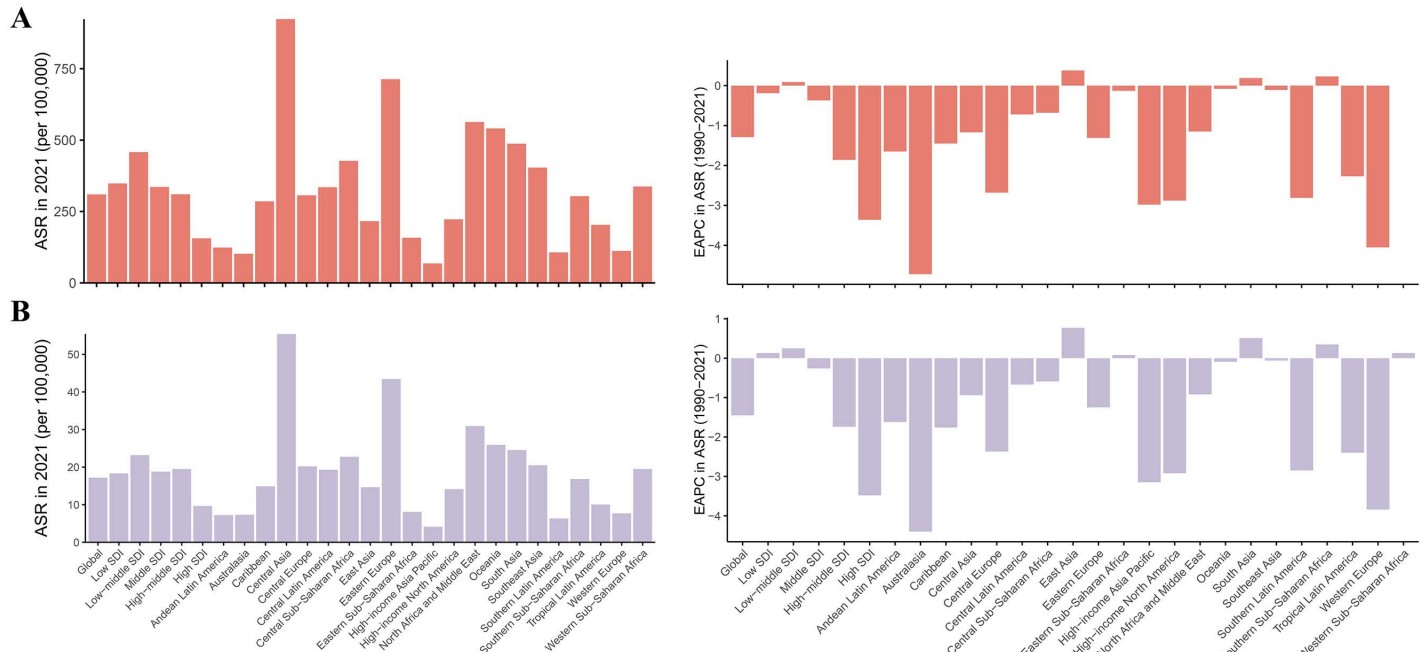

**Fig 3. Regional disparities and temporal trends in ischemic heart disease (IHD) attributable to kidney dysfunction (KD), 2021. (A)** Age-standardized DALY rates (ASDR) and estimated annual percentage changes (EAPCs) across 21 Global Burden of Disease (GBD) regions. **(B)** Age-standardized mortality rates (ASMR) and EAPCs in the same regions. Considerable regional heterogeneity was observed, with some regions showing increasing trends while others experienced substantial declines. Abbreviations: KD, kidney dysfunction; IHD, ischemic heart disease; ASDR, age-standardized DALY rate; ASMR, age-standardized mortality rate; DALYs, disability-adjusted life years; EAPC, estimated annual percentage change.

of ischemic events and premature cardiovascular death [27,28]. Behavioural and care-seeking differences (later presentation and lower adherence to preventive therapies among men) further amplify this risk, while biological factors—such as vasculo-protective effects of oestrogens versus potentially deleterious androgen-mediated pathways, and sex differences in endothelial function, oxidative stress, and immune responses—contribute to divergent outcomes [29,30]. Health-system and selection effects (for example, differential referral patterns and timing of renal-replacement therapy) may additionally modulate sex patterns across disease stages, and the male disadvantage appears to become more pronounced with advancing age [31]. Together, these epidemiologic, behavioural, biological, and system-level mechanisms form a coherent explanatory framework and underscore the need for sex-sensitive strategies—particularly earlier detection and improved control of hypertension and other cardiovascular risks in high-risk men—to reduce the disproportionate KD-related IHD burden.

Age-specific trends revealed a sharp rise in KD-attributable IHD burden among individuals aged ≥50 years, with mortality rates escalating significantly after the age of 60. This pattern is consistent with the aging-related decline in renal and cardiovascular function, which increases vulnerability to comorbid organ failure [32]. Age itself is a well-established independent risk factor for IHD mortality [33]. With advancing age, structural and functional renal changes—including glomerulosclerosis and vascular calcification—amplify susceptibility to both acute and chronic KD [23,34]. Additionally, aging is accompanied by behavioral and metabolic changes such as reduced physical activity, suboptimal nutrition, and increased frailty, all of which likely contribute to the rising IHD mortality observed in elderly populations [35–37].

## Socio-demographic disparities and regional trends

The overall increase in global DALYs and deaths due to KD-attributable IHD between 1990 and 2021 (18.7% and 22.4%, respectively) reflects the growing prevalence of CKD and its associated cardiovascular risks. However, despite the rising

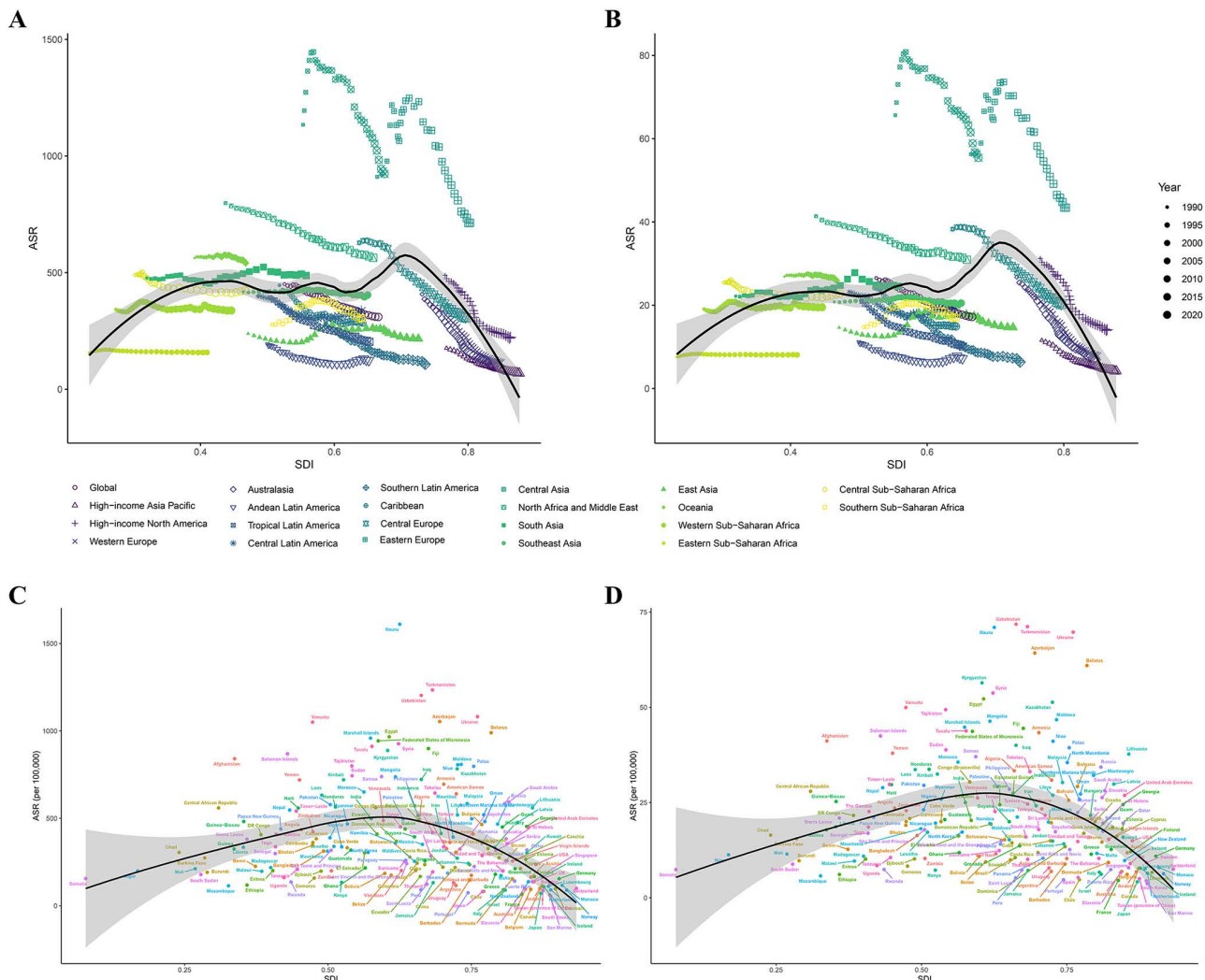

**Fig 4. Socio-demographic disparities and development thresholds in the burden of ischemic heart disease (IHD) attributable to kidney dysfunction (KD).** **(A)** Age-standardized DALY rates (ASDR) from 1990 to 2021 across 21 GBD regions, stratified by socio-demographic index (SDI). **(B)** Age-standardized mortality rates (ASMR) over the same period, also stratified by SDI region. **(C)** ASDR by SDI quintiles across 204 countries and territories in 2021. **(D)** ASMR by SDI quintiles in the same countries and year. Steeper declines were observed in high-SDI regions, while low- and middle-SDI areas showed stagnating or increasing rates. Abbreviations: KD, kidney dysfunction; IHD, ischemic heart disease; DALYs, disability-adjusted life years; ASDR, age-standardized DALY rate; ASMR, age-standardized mortality rate; SDI, socio-demographic index.

absolute burden, ASRs for both DALYs and mortality demonstrated significant declines, particularly in high-SDI regions. These reductions likely reflect improvements in healthcare access, better management of CKD and hypertension, and overall advancements in cardiovascular health. Notably, the observed decline in high-SDI countries contrasts sharply with increasing trends in low-middle SDI regions, emphasizing the disparate impacts of socio-economic development on health outcomes [38,39]. This trend suggests that while medical advancements have reduced mortality in high-income countries, low- and middle-income nations—facing greater barriers to care—continue to bear a disproportionate burden [40].

A striking finding of this study is the considerable regional variation in the burden of KD-attributable IHD. Central Asia recorded the highest age-standardized DALYs and mortality rates, a result that highlights the urgent need for improved CKD detection and management in this region. The Eastern European countries, such as Belarus, also exhibited very

high disease burden, suggesting that post-Soviet healthcare systems may still face significant challenges in addressing both cardiovascular and renal health. In contrast, high-income regions, such as High-income Asia Pacific and Scandinavia, exhibited the lowest rates, reflecting the impact of strong healthcare infrastructures, public health initiatives, and early CKD detection programs.

Interestingly, East Asia and Southern Sub-Saharan Africa exhibited rising burdens, signaling that the effects of urbanization, lifestyle changes, and aging populations may be contributing to the increased burden of KD-attributable IHD in these regions. Countries in Sub-Saharan Africa, where access to healthcare remains limited, are particularly vulnerable to the compounded effects of poverty, malnutrition, and inadequate healthcare infrastructure, further aggravating the burden of both kidney and heart diseases [41,42].

One of the most compelling findings of this study is the inverse relationship between SDI levels and KD-attributable IHD burden. Regions with higher SDI demonstrated substantial declines in age-standardized rates, reinforcing the protective role of socioeconomic development on chronic disease outcomes [43,44]. These findings align with previous research indicating that wealthier nations tend to have better healthcare systems and preventive strategies that reduce the impact of chronic diseases like IHD and CKD. Conversely, low- and low-middle SDI countries continue to face increasing burdens, likely due to fragmented health services, underdiagnosis of CKD, and limited management of common risk factors like hypertension and diabetes. Environmental stressors—such as air and water pollution—and restricted access to healthy food further exacerbate the cardiovascular risk in these regions [45]. The identification of threshold effects, with significant declines in age-standardized rates observed only in regions with SDI values above 0.7, suggests that meaningful improvements in health outcomes require more than incremental advances in healthcare systems. Instead, a more comprehensive approach addressing broader socio-economic determinants of health is necessary for substantial progress.

### Policy implications and context-specific strategies

The findings of this study carry important implications for global health planning. In low-SDI regions—such as sub-Saharan Africa—policy priorities should focus on strengthening primary care infrastructure and implementing community-based screening programs that jointly target hypertension, diabetes, and CKD. Integration with local health posts or mobile clinics can facilitate early identification of high-risk individuals. Programs such as the WHO HEARTS initiative, already piloted in several African countries, offer scalable frameworks that combine standardized treatment protocols, essential medication access, and lifestyle counseling for cardiovascular risk management. Embedding CKD screening into these existing platforms may enhance efficiency and improve coverage [46,47].

In middle-SDI settings, such as South Asia and Latin America, expanding cost-effective pilot programs that integrate diabetes and CKD screening into rural or peri-urban clinics has shown promising results. For instance, India's India Hypertension Control Initiative (IHCI) and Sri Lanka's primary care strengthening programs have demonstrated feasibility in task-shifting and protocol-based management. Scaling these models—supported by training non-physician health workers and digital health tools—can bridge resource gaps and improve continuity of care for patients with early KD [25,48,49].

In high-SDI countries, the focus should shift to integrating CKD care within existing cardiovascular disease management pathways, ensuring continuity from primary to specialist care. Additionally, addressing residual disparities—particularly among older adults and males, who face disproportionately higher burdens—requires targeted interventions, such as sex-specific health education campaigns and screening reminders integrated into electronic health records [50–52]. In addition, sodium-glucose cotransporter-2 inhibitors (SGLT2is) should be incorporated into routine treatment protocols. These medications have been shown to provide both kidney and cardiovascular protection in CKD patients and may help further reduce the disease burden in high-SDI settings [53].

Importantly, based on current epidemiological evidence, it is recommended that prevention strategies for KD-attributable IHD be embedded within broader NCD control platforms. Shared upstream risk factors—particularly elevated

systolic blood pressure, hyperglycemia, tobacco use, and unhealthy diets—must be tackled through a multisectoral approach. This includes implementing fiscal and regulatory policies such as salt reformulation, sugar taxes, front-of-package labeling, and tobacco taxation, while improving access to essential medications like antihypertensives, statins, and diabetes treatments. Such integrated policies can deliver cross-cutting cardiovascular and renal benefits, especially in resource-limited settings [54,55].

Tailoring strategies to local epidemiological trends, healthcare system capacity, and SDI level is crucial. By aligning interventions with regional needs, health systems can more effectively reduce disparities and mitigate the growing global burden of IHD attributable to KD.

While this study provides valuable insights into the global burden of KD-attributable IHD, several limitations must be acknowledged: First, although the GBD 2021 study integrates multiple data sources, including national surveys, hospital records, and cohort studies, the quality, representativeness, and completeness of data vary considerably across countries, particularly in low- and middle-income regions. Inadequate data collection systems and underreporting in resource-limited settings may introduce systematic bias into burden estimations. Second, the exclusion of individuals undergoing renal replacement therapy (dialysis or transplantation) may underestimate the disease burden in this high-risk population, who experience significant cardiovascular comorbidities. Third, while associations between socioeconomic factors, age, sex, and KD-attributable IHD burden were explored, the study's cross-sectional nature limits causal inferences. Longitudinal studies would provide clearer insights into the underlying mechanisms. Finally, While the GBD comparative risk assessment framework provides internally consistent and additive estimates across risk factors, it does not fully capture complex biological interactions between overlapping exposures. For example, hypertension and diabetes are common comorbidities in CKD patients and share mechanistic pathways that amplify IHD risk. However, their synergistic effects are not modeled jointly, potentially underestimating the true cardiovascular burden in multimorbid populations. Future studies employing mediation analysis or joint risk factor modeling could address this limitation.

## Conclusions

This study provides critical insights into the global burden of KD-attributable IHD and highlights significant geographic, sex-, and age-specific disparities. As KD continues to be a major independent risk factor for IHD, there is an urgent need for global efforts to address both conditions simultaneously. Strengthening healthcare infrastructure, particularly in low- and middle-income countries, improving KD screening, and promoting lifestyle modifications are essential steps toward reducing the global burden of cardiovascular and renal diseases.

## Supporting information

**S1 Table. Age-specific rates of DALYs and deaths for ischemic heart disease attributable to impaired kidney function in 2021.**
(XLSX)

**S2 Table. The age-standardised rates of DALYs and deaths for ischemic heart disease attributable to impaired kidney function in 204 countries and territories from 1990 to 2021.**
(XLSX)

## Acknowledgments

We truly appreciate the efforts of the Global Burden of Disease Study 2021 collaborators in delivering the most complete study of various diseases on a worldwide scale. We also express our sincere appreciation to the Institute for Health Metrics and Evaluation (IHME) for making the GBD data available for this research.

## Author contributions

**Conceptualization:** Huaipeng Zhang, Guoqing Li, Ying Zhang.

**Data curation:** Huaipeng Zhang, Guoqing Li, Ying Zhang.

**Formal analysis:** Huaipeng Zhang, Xiangbing Wang.

**Investigation:** Xiangbing Wang.

**Methodology:** Huaipeng Zhang, Guoqing Li, Xiangbing Wang, Ying Zhang.

**Project administration:** Ying Zhang.

**Software:** Huaipeng Zhang, Xiangbing Wang.

**Supervision:** Ying Zhang.

**Visualization:** Huaipeng Zhang.

**Writing – original draft:** Huaipeng Zhang, Xiangbing Wang.

**Writing – review & editing:** Guoqing Li, Ying Zhang.

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
