## [Decision Letter · Decision Letter 0]

1 Jul 2025

Dear Dr. Zhang,

Thank you for submitting your manuscript to PLOS ONE. After careful consideration, we feel that it has merit but does not fully meet PLOS ONE’s publication criteria as it currently stands. Therefore, we invite you to submit a revised version of the manuscript that addresses the points raised during the review process.

We look forward to receiving your revised manuscript.

Kind regards,

PUGAZHENTHAN THANGARAJU, M.D.,Ph.D., FRCP (LONDON)., FRCP (GLASGOW).,MBA.,

Academic Editor

PLOS ONE

Journal Requirements:

1. Please ensure that your manuscript meets PLOS ONE's style requirements, including those for file naming. The PLOS ONE style templates can be found at https://journals.plos.org/plosone/s/file?id=wjVg/PLOSOne_formatting_sample_main_body.pdf and https://journals.plos.org/plosone/s/file?id=ba62/PLOSOne_formatting_sample_title_authors_affiliations.pdf.

At this time, please upload the minimal data set necessary to replicate your study's findings to a stable, public repository (such as figshare or Dryad) and provide us with the relevant URLs, DOIs, or accession numbers that may be used to access these data. For a list of recommended repositories and additional information on PLOS standards for data deposition, please see https://journals.plos.org/plosone/s/recommended-repositories

Reviewers' comments:

Reviewer's Responses to Questions

**Comments to the Author**

1. Is the manuscript technically sound, and do the data support the conclusions?

Reviewer #1: Yes

Reviewer #2: Yes

2. Has the statistical analysis been performed appropriately and rigorously?

Reviewer #1: I Don't Know

Reviewer #2: Yes

3. Have the authors made all data underlying the findings in their manuscript fully available?

Reviewer #1: Yes

Reviewer #2: Yes

4. Is the manuscript presented in an intelligible fashion and written in standard English?

Reviewer #1: Yes

Reviewer #2: Yes

Reviewer #1: 1. Lines 48–76 (introduction) → Condense background on IHD to 2–3 concise lines and focus on the novelty of KD as an emerging factor.

2. Pages 8–15 (DALY/ASR breakdown by region) → Summarize key findings for low, middle, and high SDI groups. Include comparative statements like, “High-SDI countries saw the steepest ASR decline, while low-middle SDI groups showed slight increases.”

3. While the Discussion section recognizes disparities, the policy-level implications are generic (e.g., "integrated cardiovascular and renal care") without specifying how they may be implemented.

Suggestion:

Add region-specific recommendations (e.g., for sub-Saharan Africa: community-level hypertension screening programs linked with CKD testing).

Consider citing existing successful models (e.g., WHO HEARTS initiative) or specific public health programs.

Lines 322–334 → Expand on practical intervention models suitable for different SDI regions.

4. Switch consistently between “KD-related IHD” or “KD-attributable IHD.” Both are used interchangeably—standardize.

5. Include short summary captions for each figure (e.g., “Trends in DALYs by SDI group”).

6.

Reviewer #2: This manuscript provides a comprehensive analysis of the epidemiological trends and global disparities in the burden of ischemic heart disease (IHD) attributable to kidney dysfunction (KD). The study is timely and relevant, addressing a neglected area in cardiovascular-renal epidemiology.

But, there are certain

1. Consideration of key subpopulations such as individuals with diabetes or hypertension—known shared risk factors for CKD and IHD—could further enrich the analysis.

2. The authors could have added a brief discussion elaborating on pathophysiological mechanisms linking KD and IHD, and reviewing the evidence for specific interventions that may reduce the attributable burden (e.g., ACE inhibitors, SGLT2 inhibitors).

3.While the GBD framework offers standardized and widely accepted estimates, it is important to recognize that data quality and availability can vary substantially across regions, particularly in low- and middle-income countries. The authors could have acknowledged this limitation and consider the potential for estimation bias in these settings where health surveillance systems may be less robust.

4. The authors could have added a brief discussion on the strategies followed in the high income countries in for identifying KD.

5. A brief discussion on the policy integrated cardiovascular and renal care, particularly in resource-limited settings, which aligns with global health priorities could be better.

**Do you want your identity to be public for this peer review?** For information about this choice, including consent withdrawal, please see our Privacy Policy

Reviewer #1: **Yes: ** Sree Sudha T Y

Reviewer #2: No

---

## [Author Response · Author response to Decision Letter 1]

25 Jul 2025

Title: Global, regional, and national burden of kidney dysfunction-attributed ischemic heart disease from 1990 to 2021: a systematic analysis of the Global Burden of Disease study 2021

ID: PONE-D-25-23351

Dear Editor,

Thanks for providing us with this great opportunity to submit a revised version of our manuscript. We appreciate the detailed and constructive comments provided by the reviewers. We have carefully revised the manuscript by incorporating all the suggestions by the review panel.

In response to the comments, we have conducted a thorough literature review and detailed discussion, and revised the manuscript accordingly, with the changes highlighted in red.

We hope that our responses have adequately addressed the reviewers’ concerns and look forward to publishing our manuscript in your esteemed journal. Please do not hesitate to contact me if you require further clarification on any of the above items or have additional questions.

Kind regards,

Ying Zhang, M.D.

Department of Cardiology, Heze Municipal Hospital, Heze 274031, Shandong, China

Email: zhyinglove@163.com 

Editorial Requirements – Formatting and Data Sharing

1. Please ensure that your manuscript meets PLOS ONE's style requirements, including those for file naming

Response We standardized the format according to the templates provided in the information for authors. We have carefully revised the manuscript to ensure full compliance with the PLOS ONE formatting guidelines. This includes restructuring the manuscript sections, standardizing figure legends, and ensuring appropriate file naming conventions (e.g., Main_manuscript.docx, Supporting_information.docx, and separate high-resolution figure files). The title and author affiliations are also formatted according to the official template provided by PLOS ONE.

2. Thank you for uploading your study's underlying data set. Unfortunately, the repository you have noted in your Data Availability statement does not qualify as an acceptable data repository according to PLOS's standards. At this time, please upload the minimal data set necessary to replicate your study's findings to a stable, public repository (such as figshare or Dryad) and provide us with the relevant URLs, DOIs, or accession numbers that may be used to access these data

Response We appreciate your guidance regarding data deposition. All data used in this study are publicly available and freely accessible through the Global Burden of Disease (GBD) Results Tool provided by the Institute for Health Metrics and Evaluation (IHME). We have revised our data availability statement based on the latest publication from PLOS One (lines 405-409).

Specifically, the minimal dataset used to reproduce our analyses—including age-standardized DALYs, mortality rates, and population estimates attributable to kidney dysfunction—can be accessed at: http://ghdx.healthdata.org/gbd-results-tool. No proprietary or restricted data were used. The revised Data Availability Statement in the manuscript has been updated to reflect this (lines 405-409). If it still does not meet PLOS standards, please contact us to resolve the issue. 

Response to Reviewers

Reviewer #1:

Comment 1: Lines 48–76 (introduction) Condense background on IHD to 2–3 concise lines and focus on the novelty of KD as an emerging factor.

Response: Thank you for this suggestion. We have condensed the background on ischemic heart disease (IHD) to three concise lines, restructured the Introduction to emphasize kidney dysfunction (KD) as an emerging and independent risk factor, and highlighted the novelty of assessing KD-attributable IHD from a global and SDI-stratified perspective (lines 47-56).

Comment 2: Pages 8–15 (DALY/ASR breakdown by region) → Summarize key findings for low, middle, and high SDI groups. Include comparative statements like, “High-SDI countries saw the steepest ASR decline, while low-middle SDI groups showed slight increases.

Response: Thank you for this insightful suggestion. In response, we have added a new comparative summary paragraph to the Results section (lines 225–234), explicitly highlighting the differences in ASDR and ASMR trends across high-, middle-, and low-middle SDI regions.

Specifically, we report that high-SDI countries experienced the steepest declines in age-standardized rates (ASRs)—exceeding 60% reductions between 1990 and 2021—accompanied by the most negative EAPCs. In contrast, low-middle SDI regions showed slight but statistically significant increases in both ASDR and ASMR, with small positive EAPCs, suggesting worsening burden. Middle-SDI countries demonstrated relatively stable trends with minimal changes in ASRs.

Comment 3: While the Discussion section recognizes disparities, the policy-level implications are generic (e.g., "integrated cardiovascular and renal care") without specifying how they may be implemented. Suggestion: Add region-specific recommendations (e.g., for sub-Saharan Africa: community-level hypertension screening programs linked with CKD testing). Consider citing existing successful models (e.g., WHO HEARTS initiative) or specific public health programs. Lines 322–334 → Expand on practical intervention models suitable for different SDI regions.

Response: Thank you for this thoughtful and constructive suggestion. In response, we have substantially expanded the Discussion section (lines 327–368) to provide region-specific recommendations aligned with differing SDI contexts.

For low-SDI regions such as sub-Saharan Africa, we now recommend implementing community-level hypertension and diabetes screening programs integrated with CKD testing at the primary care level, supported by mobile clinics and task-shifting to non-physician health workers. We cite the WHO HEARTS initiative as a proven, scalable framework for integrating cardiovascular and renal care in such settings [1-3].

In middle-SDI regions, including South Asia and Latin America, we highlight the India Hypertension Control Initiative (IHCI) and Sri Lanka’s national NCD platform as effective models for embedding CKD management into broader noncommunicable disease care using protocol-based treatment and digital tools. These approaches are discussed as examples for scale-up [4-5].

In high-SDI countries, we recommend integrating CKD care into existing cardiovascular pathways, enhancing electronic health record (EHR)–based risk stratification, and addressing residual disparities through sex- and age-specific interventions [6-8]. We also emphasize incorporating SGLT2 inhibitors into standard protocols, given their proven reno- and cardioprotective effects [9].

These additions are intended to enhance the policy relevance and translational value of our findings by demonstrating how CKD-attributable IHD prevention strategies can be operationalized within diverse health system contexts.

References

[1]. Wellmann IA, Ayala LF, Valley TM, et al. Evaluating the World Health Organization's Hearts Model for Hypertension and Diabetes Management: A Pilot Implementation Study in Guatemala. Glob Heart. 2025;20(1):9. Published 2025 Jan 31. doi:10.5334/gh.1397.

[2]. Devi R, Kanitkar K, Narendhar R, Sehmi K, Subramaniam K. A Narrative Review of the Patient Journey Through the Lens of Non-communicable Diseases in Low- and Middle-Income Countries. Adv Ther. 2020;37(12):4808-4830. doi:10.1007/s12325-020-01519-3.

[3]. Al-Ghamdi S, Abu-Alfa A, Alotaibi T, et al. Chronic Kidney Disease Management in the Middle East and Africa: Concerns, Challenges, and Novel Approaches. Int J Nephrol Renovasc Dis. 2023;16:103-112. doi:10.2147/IJNRD.S363133.

[4]. Mohan V, Seedat YK, Pradeepa R. The rising burden of diabetes and hypertension in southeast asian and african regions: need for effective strategies for prevention and control in primary health care settings. Int J Hypertens. 2013;2013:409083. doi:10.1155/2013/409083.

[5]. Ameh OI, Ekrikpo UE, Kengne AP. Preventing CKD in Low- and Middle-Income Countries: A Call for Urgent Action. Kidney Int Rep. 2019;5(3):255-262. doi:10.1016/j.ekir.2019.12.013.

[6]. Narva AS, Briggs M. The National Kidney Disease Education Program: improving understanding, detection, and management of CKD. Am J Kidney Dis. 2009;53(3 Suppl 3):S115-S120. doi:10.1053/j.ajkd.2008.05.038.

[7]. German CA, Baum SJ, Ferdinand KC, et al. Defining preventive cardiology: A clinical practice statement from the American Society for Preventive Cardiology. Am J Prev Cardiol. 2022;12:100432. doi:10.1016/j.ajpc.2022.100432.

[8]. Zheng X, Yang Z, Liu S, Li Y, Wang A. Digital symptom management interventions for people with chronic kidney disease: a scoping review based on the UK Medical Research Council Framework. BMC Public Health. 2024;24(1):3534. doi:10.1186/s12889-024-20871-5.

[9]. Badve SV, Bilal A, Lee MMY, et al. Effects of GLP-1 receptor agonists on kidney and cardiovascular disease outcomes: a meta-analysis of randomised controlled trials. Lancet Diabetes Endocrinol. 2025;13(1):15-28. doi:10.1016/S2213-8587(24)00271-7.

Comment 4: Switch consistently between “KD-related IHD” or “KD-attributable IHD.” Both are used interchangeably—standardize.

Response: Thank you for pointing out this important issue. To ensure epidemiological consistency and clarity, we have standardized the terminology throughout the manuscript to “KD-attributable IHD”. This term aligns with the Global Burden of Disease (GBD) framework, which quantifies disease burden based on attributable risk estimation models. Using “attributable” more accurately reflects causal contributions from kidney dysfunction to IHD, as estimated by comparative risk assessments in the GBD methodology.

Comment 5: Include short summary captions for each figure (e.g., “Trends in DALYs by SDI group”).

Response: Thank you for this helpful suggestion. In accordance with your recommendation, we have revised the figure legends throughout the Results section to include concise and descriptive summary captions.

Each updated figure legend now begins with a brief overview sentence that summarizes the main message of the figure, followed by concise panel descriptions where applicable. We also ensured consistent definitions and expansions of abbreviations such as ASDR, ASMR, SDI, and DALYs, to improve clarity and reader accessibility. 

Reviewer #2 This manuscript provides a comprehensive analysis of the epidemiological trends and global disparities in the burden of ischemic heart disease (IHD) attributable to kidney dysfunction (KD). The study is timely and relevant, addressing a neglected area in cardiovascular-renal epidemiology. But, there are certain.

Comment 1: Consideration of key subpopulations such as individuals with diabetes or hypertension—known shared risk factors for CKD and IHD—could further enrich the analysis.

Response: Thank you for your valuable comment. We fully agree that hypertension and diabetes are important comorbidities that contribute significantly to both chronic kidney disease (CKD) and ischemic heart disease (IHD). These conditions share common pathophysiological mechanisms—such as endothelial dysfunction, systemic inflammation, and metabolic dysregulation—which may synergistically amplify cardiovascular risk, especially in populations with multiple chronic conditions.

As the reviewer rightly points out, the comparative risk assessment (CRA) framework used in the Global Burden of Disease (GBD) study estimates the burden attributable to each risk factor independently. Although this approach incorporates adjustments for confounders and comorbidities, it does not capture interaction or synergistic effects between overlapping exposures, such as concurrent kidney dysfunction and hypertension or diabetes [1, 2]. This may lead to an underestimation of the true IHD burden in multimorbid individuals.

In response to this important observation, we have made the following revisions to the manuscript:

In the Introduction, we now highlight that, unlike well-characterized metabolic risk factors such as elevated systolic blood pressure and high fasting glucose [3, 4], kidney dysfunction remains under-assessed in the context of multimorbidity (lines 50-52);

In the Limitations section, we explicitly acknowledge that the GBD framework does not account for complex interactions among risk factors, and we recommend future studies to apply joint risk modeling or mediation analysis to better understand these relationships and improve causal attribution (lines 382-389).

References

[1] GBD 2021 Risk Factors Collaborators. Global burden and strength of evidence for 88 risk factors in 204 countries and 811 subnational locations, 1990-2021: a systematic analysis for the Global Burden of Disease Study 2021. Lancet. 2024;403(10440):2162-2203. doi:10.1016/S0140-6736(24)00933-4.

[2] Safiri S, Karamzad N, Singh K, et al. Burden of ischemic heart disease and its attributable risk factors in 204 countries and territories, 1990-2019. Eur J Prev Cardiol. 2022;29(2):420-431. doi:10.1093/eurjpc/zwab213.

[3] Zhuang Z, Wang Q, Li H, Lan S, Su Y, Lin Y, Guo P: Global trends and disparities in ischemic heart disease attributable to high systolic blood pressure, 1990-2021: Insights from the global burden of disease study. PLoS One 2025, 20(6):e0324073.

[4] Wang L, Ma N, Wei L: Global burden of ischemic heart disease attributable to high sugar-sweetened beverages intake from 1990 to 2019. Nutr Metab Cardiovasc Dis 2023, 33(6):1190-1196.

Comment 2: The authors could have added a brief discussion elaborating on pathophysiological mechanisms linking KD and IHD, and reviewing the evidence for specific interventions that may reduce the attributable burden (e.g., ACE inhibitors, SGLT2 inhibitors).

Response: Thank you for this valuable suggestion. In response, we have expanded the Discussion section (lines 252-262) to summarize the principal pathophysiological mechanisms by which KD increases cardiovascular risk. These include endothelial dysfunction, chronic systemic inflammation, oxidative stress, arterial stiffness, and calcium-phosphate imbalance—all of which promote vascular calcification and accelerated coronary atherosclerosis, contributing to impaired myocardial perfusion and increased IHD burden [1,2].

Furthermore, we have included a concise summary of recent clinical evidence supporting the use of renin–angiotensin–aldosterone system (RAAS) inhibitors and sodium-glucose cotransporter 2 (SGLT2) inhibitors. These agents have been shown to reduce cardiovascular events, delay CKD progression, and lower all-cause mortality in high-risk populations, including those without diabetes [3]. Based on this evidence, we recommend their routine integration into cardiorenal treatment protocols, particularly in high-SDI settings where access to these medications is more feasible (lines 352-356).

References

[1]. Jankowski J, Floege J, Fliser D, Böhm M, Marx N: Cardiovascular Disease in Chronic Kidney Disease: Pathophysiological Insights and Therapeutic Options. Circulation 2021, 143(11):1157-1172.

[2]. Zoccali C, Mallamaci F, Adamczak M, de Oliveira RB, Massy ZA, Sarafidis P, Agarwal R, Mark PB, Kotanko P, Ferro CJ et al: Cardiovascular complications in chronic kidney disease: a review from the European Renal and Cardiovascular Medicine Working Group of the European Renal Association. Cardiovasc Res 2023, 119(11):2017-2032.

[3]. Badve SV, Bilal A, Lee MMY, et al. Effects of GLP-1 receptor agonists on kidney and cardiovascular disease outcomes: a meta-analysis of randomised controlled trials. Lancet Diabetes Endocrinol. 2025;13(1):15-28. doi:10.1016/S2213-8587(24)00271-7.

Comment 3: While the GBD framework offers standardized and widely accepted estimates, it is important to recognize that data quality and availability can vary substantially across regions, particularly in low- and middle-income countries. The authors could have acknowledged this limitation and consider the potential for estimation bias in these settings where health surveillance systems may be less robust.

Response: This is an important point. While the GBD framework offers a standardized and internally consistent approach to estimating disease burden across regions, the accuracy and precision of these estimates are inherently influenced by the quality, availability, and representativeness of primar

---

## [Decision Letter · Decision Letter 1]

28 Sep 2025

Dear Dr. Zhang,

Thank you for submitting your manuscript to PLOS ONE. After careful consideration, we feel that it has merit but does not fully meet PLOS ONE’s publication criteria as it currently stands. Therefore, we invite you to submit a revised version of the manuscript that addresses the points raised during the review process.

**ACADEMIC EDITOR: Thank you for addressing all comments from reviewers. The manuscript has improved significantly. However, an additional minor revision might be helpful. My comments are included as below.**

We look forward to receiving your revised manuscript.

Kind regards,

Thien Tan Tri Tai Truyen, M.D.

Academic Editor

PLOS ONE

Journal Requirements:

Additional Editor Comments;

1. Introduction: To improve the context, I suggest the authors using/updating this section with recent studies which have emphasized the association of CKD in both moderate and severe stages with adverse cardiovascular events (DOI: 10.1161/JAHA.125.042307 ; DOI: 10.34067/KID.0000000705 )

2. Methodology:

2.1 A brief discussion about data collecting protocol specifying which keywords and how did you collect the data might be useful for readers and other scientists to replicate similar studies in the future. 

2.2 Please also specify the formula of your parameters such as ASR, EAPC. Moreover, I believe most of data from GBD provide you 95% of Uncertainty Interval which is quite different than Confidence Interval. Please also explain this estimate. 

3 Results/Discussion

3.1 The observed sex differences are interesting and consistent with previous studies. Existing evidence indicates that the prevalence of CKD is higher in females than in males, likely due to longer life expectancy and potential overdiagnosis (DOI: 10.1038/nrneph.2017.181). However, males carry a substantially greater risk of adverse cardiovascular events. Your findings align well with this established knowledge. I recommend expanding the discussion (lines 261–271) to highlight the higher cardiovascular risk in males and provide additional rationale for this observation. This represents one of the major findings of your study and warrants more in-depth discussion.

Reviewers' comments:

Reviewer's Responses to Questions

**Comments to the Author**

Reviewer #1: All comments have been addressed

Reviewer #2: All comments have been addressed

2. Is the manuscript technically sound, and do the data support the conclusions?

Reviewer #1: Yes

Reviewer #2: Yes

3. Has the statistical analysis been performed appropriately and rigorously?

Reviewer #1: Yes

Reviewer #2: Yes

4. Have the authors made all data underlying the findings in their manuscript fully available?

Reviewer #1: Yes

Reviewer #2: Yes

5. Is the manuscript presented in an intelligible fashion and written in standard English?

Reviewer #1: Yes

Reviewer #2: Yes

Reviewer #1: Authors have condensed the introduction, standardized terminology (KD-attributable IHD), added region-specific policy recommendations (including WHO HEARTS, IHCI), and improved figure legends as per reviewer suggestions.

Reviewer #2: The authors have addressed all the queries in detail with references. The manuscript can be accepted for publication.

**Do you want your identity to be public for this peer review?** For information about this choice, including consent withdrawal, please see our Privacy Policy

Reviewer #1: **Yes: ** Dr Sree Sudha T Y

Reviewer #2: **Yes: ** Dr Meenalotchini Prakash Gurunthalingam

---

## [Author Response · Author response to Decision Letter 2]

9 Oct 2025

Title: Global, regional, and national burden of kidney dysfunction-attributed ischemic heart disease from 1990 to 2021: a systematic analysis of the Global Burden of Disease study 2021

ID: PONE-D-25-23351R1

Dear Editor,

Thanks for providing us with this great opportunity to submit a revised version of our manuscript. We appreciate the detailed and constructive comments provided by the reviewers. We have carefully revised the manuscript by incorporating all the suggestions by the review panel.

In response to the comments, we have conducted a thorough literature review and detailed discussion, and revised the manuscript accordingly, with the changes highlighted in red.

We hope that our responses have adequately addressed your concerns and look forward to publishing our manuscript in your esteemed journal. Please do not hesitate to contact me if you require further clarification on any of the above items or have additional questions.

Kind regards,

Ying Zhang, M.D.

Department of Cardiology, Heze Municipal Hospital, Heze 274031, Shandong, China

Email: zhyinglove@163.com 

Journal Requirements:

Response The reviewers' comments did not include suggestions for citing specific previously published works.

Response We have checked the reference list to ensure it is complete and accurate. We have not cited any retracted papers. 

Editor Comments

1. Introduction: To improve the context, I suggest the authors using/updating this section with recent studies which have emphasized the association of CKD in both moderate and severe stages with adverse cardiovascular events (DOI: 10.1161/JAHA.125.042307 ; DOI: 10.34067/KID.0000000705 )

Response Thank you for sharing. To improve the context, we revised the introduction by referring to these two papers (DOI: 10.1161/JAHA.125.042307; DOI: 10.34067/KID.0000000705) (lines57-63).

2. Methodology:

2.1 A brief discussion about data collecting protocol specifying which keywords and how did you collect the data might be useful for readers and other scientists to replicate similar studies in the future.

Response Thank you for your suggestion. We have added this clarification to the Methods section (lines 81-85).

2.2 Please also specify the formula of your parameters such as ASR, EAPC. Moreover, I believe most of data from GBD provide you 95% of Uncertainty Interval which is quite different than Confidence Interval. Please also explain this estimate.

Response Thank you for your valuable feedback. As requested, we would like to clarify the details regarding the Age-Standardized Rate (ASR) calculation. The ASR was provided directly from the Global Health Data Exchange (GHDx), as part of the data from the Global Burden of Disease (GBD) Study 2021. We did not perform the calculation of ASR ourselves, but rather obtained the pre-calculated rates from the GBD database, which uses the method outlined in their official documentation. The ASR formula provided by GHDx see the uploaded Word document of Response Letter.

The EAPC calculation formula has been supplemented in the statistical analysis section (lines 136-140).

You are correct that GBD data provides Uncertainty Intervals (UI) instead of Confidence Intervals (CI), and we acknowledge the difference. All GBD data, including the absolute numbers, rates, and ASRs of DALYs and deaths, provide 95% UI. Our calculations based on the data provided by GBD include 95% CI. We have supplemented the explanation of the 95% UI in the descriptive analysis section (lines 115-123).

Uncertainty Intervals (UI) reflect the range within which the true value is expected to lie, incorporating not only statistical variability (such as sampling error) but also the systematic uncertainties in the model assumptions, data sources, and input variables. The GBD study utilizes a probabilistic model to estimate health outcomes, and the UI accounts for all potential sources of uncertainty, including variations in the input data, modeling techniques, and the assumptions made during the estimation process. The 95% UI thus represents a range of possible values for a given health estimate, with 95% of possible estimates expected to fall within this interval.

Confidence Intervals (CI), by contrast, are generally derived from statistical sampling theory and primarily reflect the uncertainty due to random sampling error. A 95% CI indicates that if the same study were repeated multiple times, the true value would fall within the interval 95% of the time. However, CI does not account for the systematic uncertainties in the data or the modeling process.

Given the complexity of global health data, which integrates multiple sources and modeling approaches, Uncertainty Intervals offer a more comprehensive understanding of the variability in our estimates. By including both statistical and model-related uncertainties, 95% UIs provide a more realistic range of possible outcomes, particularly for complex epidemiological estimates like those derived from the GBD study.

3 Results/Discussion

3.1 The observed sex differences are interesting and consistent with previous studies. Existing evidence indicates that the prevalence of CKD is higher in females than in males, likely due to longer life expectancy and potential overdiagnosis (DOI: 10.1038/nrneph.2017.181). However, males carry a substantially greater risk of adverse cardiovascular events. Your findings align well with this established knowledge. I recommend expanding the discussion (lines 261–271) to highlight the higher cardiovascular risk in males and provide additional rationale for this observation. This represents one of the major findings of your study and warrants more in-depth discussion.

Response Thank you for your insightful feedback and suggestions. In response to your recommendation, we have expanded the discussion in lines 282–303 to highlight the higher cardiovascular risk observed in males despite the higher prevalence of chronic kidney disease (CKD) in females. We have provided additional rationale to explain this finding, based on several key factors that contribute to the observed disparity:

1. Differences in CKD Prevalence: We acknowledge that women have a higher prevalence of CKD, which is partly influenced by demographic factors such as longer life expectancy and the overestimation of CKD in low-muscle-mass individuals due to creatinine-based eGFR equations.

2. Faster Progression in Men: Men tend to progress more rapidly to advanced CKD and are more likely to accumulate cardiometabolic comorbidities, such as poorly controlled hypertension, diabetes, and smoking. These factors significantly increase their risk of ischemic events and premature cardiovascular death.

3. Behavioral and Care-Seeking Differences: Men also tend to present later for medical care and are less likely to adhere to preventive therapies, which further exacerbates their cardiovascular risk.

4. Biological Differences: We expanded on the biological mechanisms at play, including the vasculoprotective effects of estrogen in women versus the potentially harmful effects of androgens, as well as sex differences in endothelial function, oxidative stress, and immune responses.

5. Health-System and Selection Effects: Additionally, we discussed how differential referral patterns and the timing of renal-replacement therapy can influence outcomes, with the male disadvantage becoming more pronounced as patients age.

We believe that these expanded points better support the observed findings and provide a more comprehensive understanding of the higher cardiovascular risk in males with kidney dysfunction. We have incorporated these ideas to further emphasize the need for sex-sensitive strategies in managing kidney disease and ischemic heart disease, especially in high-risk men.

Thank you again for your valuable input. We trust that these revisions strengthen the manuscript and address your concerns.

---

## [Editor Report · Decision Letter 2]

14 Oct 2025

Global, regional, and national burden of kidney dysfunction-attributed ischemic heart disease from 1990 to 2021: a systematic analysis of the Global Burden of Disease study 2021

PONE-D-25-23351R2

Dear Dr. Zhang,

We’re pleased to inform you that your manuscript has been judged scientifically suitable for publication and will be formally accepted for publication once it meets all outstanding technical requirements.

Kind regards,

Thien Tan Tri Tai Truyen, M.D.

Academic Editor

PLOS ONE
---

## [Editor Report · Acceptance letter]

PONE-D-25-23351R2

PLOS One

Dear Dr. Zhang,

I'm pleased to inform you that your manuscript has been deemed suitable for publication in PLOS One. Congratulations! Your manuscript is now being handed over to our production team.

Kind regards,

on behalf of

Dr. Thien Tan Tri Tai Truyen

Academic Editor

PLOS One